# *BumbleBee*: Dynamic KV-Cache Streaming Submodular Summarization for Infinite-Context Transformers

**Lilly Kumari**$^{\diamond\,\ddagger}$     **Shengjie Wang**$^{\diamond\,\dagger}$     **Tianyi Zhou**$^{\diamond\,\star}$     **Nikhil Sarda**$^{\ddagger}$
**Anthony Rowe**$^{\S}$     **Jeff Bilmes**$^{\diamond}$
$^{\diamond}$University of Washington, Seattle, $^{\dagger}$NYU Shanghai, $^{\star}$University of Maryland,
$^{\ddagger}$Google, $^{\S}$Carnegie Mellon University
{lkumari, bilmes}@uw.edu

## Abstract

The need for Transformer-based Large Language Models (LLMs) to maintain key-value representations (a KV cache) of previously seen tokens in the GPU memory leads to a significant overhead that scales linearly with the sequence length and batch size. With the advent of extremely long context LLMs, efficiently modeling long-range dependencies becomes challenging. In this work, we focus on the problem of long context summarization by formulating it as a subset selection problem. Specifically, we propose a novel submodular optimization framework called *BumbleBee* that uses a mixture of submodular functions to balance the diversity amongst the context tokens in the key embedding space and their importance computed using accumulated attention attributed to them across different input tokens. Our framework can work for both the LLM prefill and decoding phases, utilizing offline or online versions of our submodular algorithm respectively. While the context sizes grow to be as large only as the summary size, the temporal extent of the contexts may grow unboundedly, justifying the moniker "Infinite-Context Transformers." Empirically, we validate the effectiveness of our framework across 13 different datasets using the LLaMA 7B and 13B models. Our results show that *BumbleBee* improves accuracy compared to state-of-the-art techniques at comparable context reduction ratios.

## 1 Introduction

The multi-headed self-attention (Vaswani et al., 2017) serves as the building block of several state-of-the-art transformer-based models for various tasks such as language understanding and generation (Raffel et al., 2020), image recognition (Dosovitskiy et al., 2021), and recommendations (Wu et al., 2023) using Large Language Models (LLMs). It enables models to learn long-range dependencies and complex patterns by focusing on multiple sequence sections simultaneously, regardless of their proximity. However, deploying LLMs is challenging for at least two main reasons: (1) quadratic scaling of attention: the attention mechanism scales quadratically with sequence length, leading to high computational costs and memory requirements for processing longer sequences; (2) autoregressive decoding wherein the LLM generates the output sequence token by token conditioned on the previously generated results and the input sequence, which requires accessing or recomputing the key and value representation of all previous tokens.

Existing approaches to tackle the first problem include input sequence truncation (Devlin et al., 2018; Lewis et al., 2019), sliding window-based sequence chunking (Li et al., 2020; Gong et al., 2020), memory augmentation methods (Dai et al., 2019; Rae et al., 2019; Martins et al., 2021; Zemlyanskiy et al., 2021), and retrieval-based methods (Khandelwal et al., 2019; Wu et al., 2022; Borgeaud et al., 2022; Zhong et al., 2022). However, conventional techniques like chunking suffer from well-known problems such as context fragmentation (Dai et al., 2019). Similarly, retrieval-based non-parametric methods to augment an input query with relevant context require constructing an external memory bank that is much larger than the

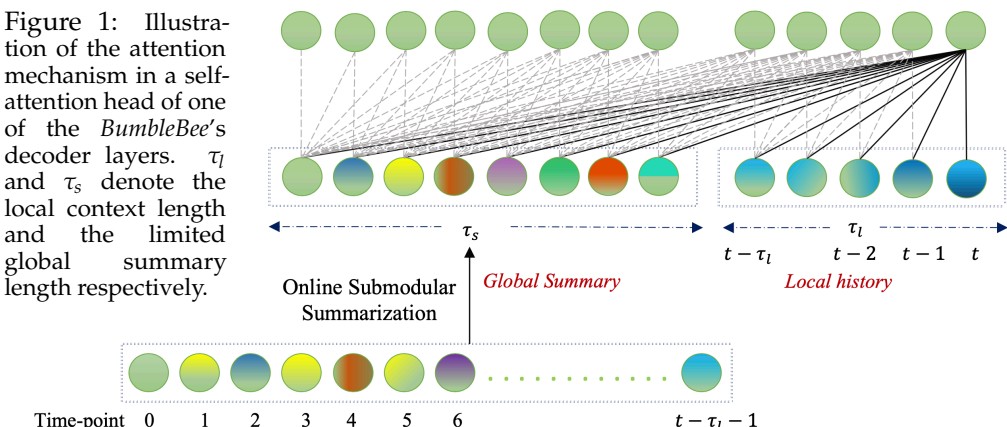

Figure 1: Illustration of the attention mechanism in a self-attention head of one of the *BumbleBee*'s decoder layers. $\tau_l$ and $\tau_s$ denote the local context length and the limited global summary length respectively.

dataset size. In addition, a retriever index must be constructed on top of the memory bank.

A common approach to address the second problem is to store the previously generated key and value vectors in a KV cache (Pope et al., 2023) which helps avoid redundant computation at each generation step. This presents a memory bottleneck, as the KV cache size grows linearly with the generated sequence length and the batch size for a specific model. Additionally, as model size increases, the memory footprint of a single token increases as well. While it is possible to offload the KV cache to the host DRAM, this will incur a host-to-device latency for each inference call. Given the growth in model size and context lengths of publicly available models (Reid et al., 2024; Anthropic, a), reducing the memory overhead of the KV cache while retaining model accuracy becomes even more important.

Existing system-level techniques such as FlexGen (Sheng et al., 2023), PagedAttention (Kwon et al., 2023), and FlashAttention (Dao et al., 2022) improve the utilization of GPU resources and throughput when dealing with the attention mechanism and KV caches. However, they do not consider the impact of ever-increasing KV cache sizes. Modeling techniques such as multi-query (Shazeer, 2019) and group-query attention (Ainslie et al., 2023) help mitigate the size of the KV cache by removing unnecessary heads but they require expensive retraining/fine-tuning.

Given the wide deployment of accelerators and models already in the field, we need to leverage inference time techniques that reduce the size of the available context at all layers (KV cache) without impacting accuracy. Existing related works such as Heavy Hitters (Zhang et al., 2023), Scissorhands (Liu et al., 2023), KeyFormer (Adnan et al., 2024) and FastGen (Ge et al., 2023) rely on heuristic-based techniques to preserve only the important key-value attention states while evicting the non-important attention states. They employ **modular scores** such as attention scores attributed to different tokens accumulated over different time steps. This does not take into account the importance of keeping a particular KV attention state in the context of other states already present in the KV cache and thus does not efficiently capture long-term dependencies between tokens. We hypothesize that KV cache selection should be framed as a subset selection problem where we evaluate the utility of different key-value pairs as a set instead of independently evaluating each key-value attention state.

In this work, we present a novel yet simple approach called **BumbleBee**[1] for KV cache summarization, thus enabling existing LLMs to be used for longer contexts without any additional fine-tuning, CPU offloading (Sheng et al., 2023), or context truncation (Beltagy et al., 2020) in case of longer sequences (Fig. 1). The temporal span of the context (which is defined as the difference between the time of the latest and the time of the earliest context token) may be unboundedly long in a *BumbleBee* model. This is true even though the total number of tokens in a *BumbleBee* context does not grow unboundedly. Indeed, *BumbleBee* draws inspiration from the following aspects of human psychology:

---

[1]Named after our favorite transformer who is known for being "efficient", highly "acute", and "adaptable".

1. Selective attention (Chun & Turk-Browne, 2007; Uncapher & Rugg, 2009) allows us to focus on relevant information and filter out distractions or irrelevant details. This plays a critical part in determining what subset of information should get successfully encoded into our memory. Similarly, in *BumbleBee*, we maintain a memory of diverse, representative, and important time points of the sequence observed so far by utilizing online submodular summarization.

2. Humans process information in an online and dynamic fashion over an extended period, possibly all the way to very early in life. This involves relying on our (possibly extremely long) past memory to make sense of incoming information through associations, selectively attending to relevant information, and organizing our existing memory in light of new information. This top-down processing style (Gazzaley & Nobre, 2012) enables us to prioritize and encode pertinent information for future recall. Similarly, in the case of *BumbleBee*, we process an incoming segment or chunk by contextualizing it with the online summary (or memory) obtained thus far and update the summary/memory based on the segment's informativeness and its association with the current memory. Like human memory, *BumbleBee*'s online summary may include tokens anywhere from the earliest point in the stream all the way to the present.

To realize the above inspiration, *BumbleBee* utilizes online submodular summarization to maintain a diverse subset of important key-value attention states on the fly that are representative of long-term global history. Incorporating a fixed-size summary of the entire historical sequence allows the *BumbleBee* to capture dependencies having arbitrarily long range without also unboundedly increasing the memory consumption or the length of input fed to the transformer. That is, the memory grows no larger than the size of the summaries. However, since the temporal extent itself between the oldest and newest token present in the summary-sized context windows can grow unboundedly, BumbleBee deservedly may be seen as an instance of an 'Infinite-Context" transformer. We further combine the global submodular summary with the most recent local context as shown in Fig. 1, so *BumbleBee*'s predictions are dependent on both the latest context as well as the global wide-ranging patterns observed so far in the temporal history. Thus, our approach boosts the transformer's capabilities to capture both the arbitrary-long coarse-grained long-term and the detailed short-term contextual dependencies while maintaining similar (and thus feasible) memory and computational costs for a fixed-size input sequence length.

## 2 Motivation

LLMs are pre-trained using a fixed context window and prior work has shown their limited generalization on sequences significantly longer than the pre-trained context window (Kazemnejad et al., 2024). Certain recent closed sourced models have a context window longer than 128k tokens, for example, GPT-4 (OpenAI, 2023) has a context window of 128k, Claude 2.1 (Anthropic, b) can process 200k tokens while Gemini 1.5 pro (Reid et al., 2024) can handle a 1M input context. To utilize such a long context, caching of key-value states is utilized to keep the latency associated with the inference phase low. However, the KV cache grows linearly with the sequence length. For example, for a LLaMA-13B model (Touvron et al., 2023b), the KV cache for 128k tokens would roughly require 40 (num. of decoder layers) $\times$5120 (hidden size) $\times$2 (kv pair) $\times$2 (fp16) $\times$128000 = 105 GB.

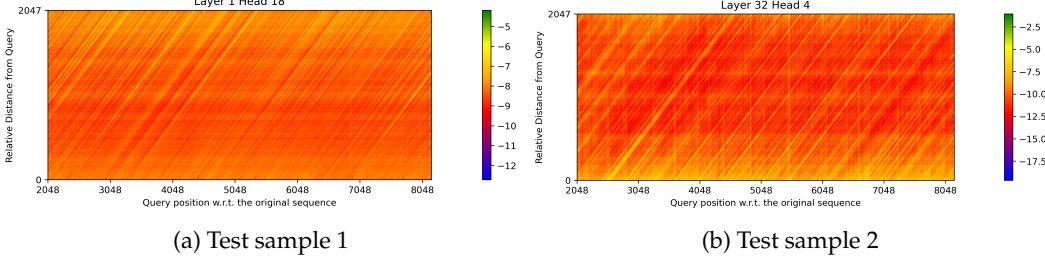

(a) Test sample 1          (b) Test sample 2

Figure 2: Attention maps for two different WikiText-103 articles using LLaMA-7B model.

To address the memory costs associated with the KV cache, we focus on the following:

*Is it possible to maintain the LLM's performance on downstream tasks without storing every observed token representation in the KV cache?*

We conduct experiments on the WikiText-103 (Merity et al., 2016) dataset to support our hypothesis that self-attention is selective and keeping only a set of **both** diverse and important tokens is sufficient to maintain performance. Specifically, we use a LLaMA-7B model (with a pre-trained context window size of 2048) for a next-token prediction task on randomly sampled articles from wikitext-103 and visualize the attention scores attributed to different tokens present in the context window of size 2048 in Figure 2. On the x-axis, each token represents the query used for next-token prediction, while each vertical slice (column) illustrates the logarithm of normalized attention scores for the 2048 in-context tokens (or keys). As we move from the bottom row to the topmost row along the y-axis, the relative distance of the in-context keys from the query token increases from 0 to 2047. The anti-diagonal pattern shown in Figure 2 shows that there is a small subset of tokens that are strongly attended to while they are present in the context window.

However, the constraint of the context window restricts tokens in longer sequences to only attend to nearby local tokens, limiting their ability to capture broader contextual information. In this work, we show that by using a submodular function to summarize the entire historical contextual information in the KV cache across different attention heads, one can reduce the memory footprint associated with the KV cache while maintaining acceptable performance compared to using the entire cache.

## 3   Related Work

**Attention speedup**: Self-attention is a critical component of the transformer (Vaswani et al., 2017) mechanism powering modern language models but suffers from quadratic complexity. Techniques such as Linformer (Wang et al., 2020), Performer (Choromanski et al., 2020), Linear Transformers (Katharopoulos et al., 2020) and Reformer (Kitaev et al., 2020) aim to reduce the time complexity of self-attention via low-rank approximations or hashing techniques. Longformer (Beltagy et al., 2020) introduces sliding-window attention to reduce the computational overhead of self-attention. FlashAttention (Dao et al., 2022) implements an IO-aware kernel to compute self-attention and uses tiling to reduce memory overhead. Keyformer (Adnan et al., 2024) speeds up attention computation by exploiting the empirical observation that 90% of attention weights focus on a small subset of tokens. For workloads that share similar inputs, prompt caching (Gim et al., 2023) and prefix sharing (Ye et al., 2024) have emerged to reduce the computation and memory overhead of self-attention.

**KV cache compression and management**: H2O (Zhang et al., 2023) identifies "heavy-hitter tokens" via a modular score function (i.e., simply the sum of individual token scores) and keeps those while evicting the rest, reducing the size of the KV cache. In Mu et al. (2024), prompts are compressed to gist tokens to reduce the size of the KV cache. FastGen (Ge et al., 2023) uses profiling information from the attention computation to determine which tokens to evict from the KV cache.

Additionally, several post-training quantization techniques have been proposed to reduce the memory overhead of the KV cache. WKVQuant (Yue et al., 2024) proposes a past-only quantization mechanism to enable higher precision for self-attention computation. In Yang et al. (2024), important tokens are retained at full precision while the rest are quantized. Quality-aware Quantization (Dong et al., 2024) uses an attention-aware approach to selectively quantize elements in the KV cache. GEAR (Kang et al., 2024) approximates quantization error via a low-rank matrix and uses a sparse matrix to correct them at inference. Sparq (Ribar et al., 2024) drops tokens according to attention sparsity scores and incorporates the error of the pruned value cache. Unlike the lossy compression techniques above, Loma (Wang & Xiao, 2024) proposes a lossless method for compressing the KV cache. However, it requires further fine-tuning of the underlying model.

Finally, efficient management of the KV cache, such as PagedAttention (Kwon et al., 2023) improves the throughput of LLM serving by reducing fragmentation and redundant dupli-

cation.

**Applications of submodularity to LLMs**: INGENIOUS (Renduchintala et al., 2023) is a technique that uses submodular optimization for selecting representative subsets of the training data such that the language models trained thereof achieve comparable performance to models trained on the full dataset. SMART (Renduchintala et al., 2024) proposes a data mixture strategy for instruction tuning, leveraging a submodular function for importance score assignment to tasks that are used to determine the mixture weights. Submodular functions have also been used to augment LLMs for multi-document summarization (Kurisinkel & Chen, 2023). Div-S3 (Kumari et al., 2024) summarizes via the submodular span (Kumari & Bilmes, 2021) based on the conditional submodular gain function. This approach facilitates the selection of diverse and relevant exemplars for in-context learning with LLMs in a data-efficient manner. To the best of our knowledge, ours is the first work to apply submodularity for KV cache summarization.

## 4 Background

In this section, we provide a concise background about self-attention used in transformer-based LLMs. We also briefly discuss submodular function optimization as well as the KV cache mechanism used to avoid re-computations during the decoding stage.

**Notation:** We denote a sequence of tokens as $x = \{x_1, x_2, \ldots x_n\}$ where $x_t \in \mathbf{R}^d$ is the $t$-th token embedding. In a decoder-only transformer model, we denote the number of decoder layers present as $n_l$ and the number of attention heads present in each layer as $n_h$. The self-attention mechanism of head $h$ in layer $(l + 1)$, utilizing distinct weight matrices for query ($W_q$), key ($W_k$), and value ($W_v$), operates on the hidden states from the preceding layer, represented as $H^l$. Specifically, we get the following query, key, and value embeddings after the linear projections: $Q^{l+1} = H^l W_q$, $K^{l+1} = H^l W_k$, and $V^{l+1} = H^l W_v$. Row $t$ in the key embedding matrix denotes the key vector corresponding to the $t$-th token, and this holds for both query and value matrices as well. Given query $q_t$, its attention output $o_t$ using scaled dot product attention is computed as shown in Eq. 1. Here, we use $S_t$ to denote the KV cache accumulated at step $t$. From here onward, we omit notation pertaining to the specific layer and head for the sake of simplicity.

$$o_t = \sum_{(k_j, v_j) \in S_t} \boxed{\frac{\exp\left(k_j^T q_t\right)}{\sum_{k_i \in S_t} \exp\left(k_i^T q_t\right)}} v_j = \sum_{(k_j, v_j) \in S_t} \boxed{a(q_t, k_j, S_t)} v_j \tag{1}$$

**KV Cache:** Decoder-only transformers operate in an autoregressive manner, predicting tokens sequentially based on previously generated (and observed) ones. By caching previously computed representations, specifically the embeddings for keys and values corresponding to observed tokens, the model can avoid redundant computations in each decoding step. Specifically, the KV cache initially computes the attention states for an input prompt, represented by $S_0 = \{(k_i, v_i) | i \leq n\}$, and caches them in memory. For every subsequent step $j \leq k$, the model reuses the cached states $S_j = \{(k_i, v_i) | i < n + j\}$ to compute the attention state $(k_{n+j}, v_{n+j})$ of the new token $s_{n+j}$. This significantly reduces the floating-point computations used for matrix operations to compute the new attention states. After each step, the newly computed attention states are appended to the cache for subsequent use, such that $S_j = S_{j-1} \cup \{(k_{n+j}, v_{n+j})\}$.

Note that the KV cache is an approximation and trades off precision for speed. The attention state computation for token $s_j$ is limited to the sequence available at step $j$, namely $\{s_i | i < n + j\}$ as opposed to over the entire sequence $\{s_i | i < n + k\}$. KV caches significantly reduce latency and result in a minimal loss in accuracy making it an essential ingredient in practical LLM deployment.

Since the KV cache is updated with a new token at each decoding step, its size grows linearly with the overall sequence length (including both input prompt and generated tokens) and the batch size. This becomes a major bottleneck when dealing with longer sequences whose

KV cache cannot fit in the GPU's high bandwidth memory (HBM). In this work, therefore, we study how to summarize the KV cache by keeping only a small subset of important and diverse and thus representative key-value embeddings while discarding the rest.

**Submodularity:** A submodular (Bilmes et al., 2022) function $f : 2^V \rightarrow \mathcal{R}$ defined on the ground set $V$ always has a diminishing return property: $f(v|A) \geq f(v|B)$ for any $v \notin B$ and $A \subseteq B \subseteq V$, where $f(v|A) := f(A \cup \{v\}) - f(A)$. Intuitively, the gain of an item $v$, i.e., $f(v|A)$, diminishes as the conditioning set grows from $A$ to $B$. A submodular valuation $f(A)$ expresses the diversity and representativeness of the input set $A \subseteq V$ of items. There are many useful such functions, one of them being the facility location (FL) function which is similar to a k-medoids objective (Kaufman & Rousseeuw, 1987). The FL function utilizes similarity scores $\text{sim}(v, v')$ computed over every pair $v, v' \in V$ of items. A valuation is then the sum of similarities from any item in the ground set $V$ to its closest representative in the given set $A$ as shown in Eq. (2).

$$f_{\text{FL}}(A) = \sum_{v \in V} \max_{v' \in A} \text{sim}(v, v'). \quad (2)$$

$$c(A) = \sum_{u \in U} \phi_u \left( \sum_{v \in A} m_u(v) \right). \quad (3)$$

A subset $A$ with a high function value indicates that for every item in the ground set, there exists an item in $A$ that is very similar, or in other words, $A$ is representative of the ground set $V$. Another widely used submodular function is the feature-based function, and it has the form shown in Eq. (3). Here, we have $\phi_u(\cdot)$ as a monotone non-decreasing non-negative concave function, and $m_u(\cdot)$ is a non-negative weight associated with the $u$-th feature of every item $v \in A$. Due to the diminishing property of $\phi_u(\cdot)$, to have a large function valuation for a set $A$, we would require the sum of every feature across items to be uniformly large, thus inducing diversity and fairness over a feature representation of the selected subset.

## 5 BumbleBee

In Section 2, we saw that some keys despite their distance from the query are heavily attended to, showing that it is important to preserve these keys in the overall context even if they are quite distant. While the importance of keys captured by the attention scores is one aspect that should be considered for context (or KV cache) selection, the diversity and representativeness of the selected keys are equally critical. That is, we require the context summary (which has a fixed size) to be both relevant to and representative of the entire context. This is how efficiency is achieved — amongst sets of a given fixed size, a diverse set means a non-redundant set, while a redundant set means that certain concepts are inefficiently over-represented while some concepts are poorly represented. To capture these properties (diversity and importance), our final scoring function $g_\lambda$ is a convex combination

---

**Algorithm 1** Offline Submodular KV cache Summarization during Prefill/Encoding Phase

---

1: **Input:** Submodular functions capturing diversity $f_{\text{FL}}$ in the key embeddings space and importance $c$ via attention frequency for layer $l$ and attention head $h$; mixture function $g_\lambda(\cdot) = \lambda f_{\text{FL}}(\cdot) + (1 - \lambda)c(\cdot)$; a set of $n$ KV attention states $K_n = \{(k_i)\}_{i=1}^n$, $V_n = \{(v_i)\}_{i=1}^n$ corresponding to the $n$ prompt tokens; budget $\tau_s$.

2: **Output:** A final summary $S_n$ such that $S_n \subseteq \{(k_i, v_i)\}_{i=1}^n$ and $|S_n| \leq \tau_s$.

3: **Initialize:** $S_n = \emptyset$; compute accumulated attention score vectors $a_n$ for each key $k \in \{k_i\}_{i=1}^n$. $a_n^i$ denotes accumulated attention scores attributed to key $k_i$ across all $n$ query tokens.

4: **for** $j = 1$ to $\tau_s$ **do**

5: $\quad k_{\text{imp}} \leftarrow \text{argmax}_{e \in K_n \setminus S_n} g_\lambda(S_n \cup e) - g(S_n)$

6: $\quad S_n \leftarrow S_n \cup \{(k_{\text{imp}}, v_{\text{imp}})\}$ where $v_{\text{imp}}$ is the value embedding associated with $k_{\text{imp}}$.

7: **end for**

---

---

**Algorithm 2** *BumbleBee*: Streaming Submodular KV cache Summarization for Transformers

---

1: **Input:** Submodular functions for diversity $f_{FL}$ in the key embeddings space and importance $c$ w.r.t. attention frequency resp. for layer $l$ and attention head $h$; mixture function $g_\lambda(\cdot) = \lambda f_{FL}(\cdot) + (1 - \lambda)c(\cdot)$; stream of QKV attention states $\{(q_i, k_i, v_i)\}_{i=1}^n$; budget $\tau_s$.
2: **Output:** A running summary $S_t$ of for every time step $t$ such that $S_t \subseteq \{(k_i, v_i)\}_{i=1}^t$.
3: **Initialize:** $S_0 = \varnothing$, $a_0 = \varnothing$ where $a_t \in \mathbf{R}^{|S_t|}$ denotes the accumulated attention scores corresponding to keys present in $S_t$ across $t$ time steps.
4: **for** $t = 1, \dots, n$ **do**
5:  Update $a_t$ for each $k \in S_{t-1}$ by adding $a(q_t, k, S_{t-1} \cup k_t)$
6:  **if** $t < \tau_s$ **then**
7:   $S_t \leftarrow S_{t-1} \cup \{(k_t, v_t)\}$
8:   Append $a(q_t, k_t, S_t)$ to $a_t$ s.t. $|a_t| = |S_t|$
9:  **else**
10:   Let $S'_t = S_{t-1} \cup \{(k_t, v_t)\}$;  $k_{discard} \leftarrow \operatorname{argmin}_{k_i \in S'_t} g_\lambda(k_i | S'_t \setminus k_i)$
11:   $S_t \leftarrow S'_t \setminus \{(k_{discard}, v_{discard})\}$
12:   **if** $k_{discard} \neq k_t$ **then**
13:    Evict $a_t^j$ (the accumulated attention score for the discarded key $k_{discard}$) from $a_t$.
14:    Append $a(q_t, k_t, S_t)$ to $a_t$
15:   **end if**
16:  **end if**
17: **end for**

---

of the facility location (Eq. 2) and the feature-based (Eq. 3) functions.

$$g_\lambda(A) = \lambda f_{FL}(A) + (1 - \lambda)c(A) \tag{4}$$

Here, $\lambda \geq 0$ is a hyperparameter that controls the trade-off between representativeness and relevance. Both component functions are monotone, non-negative, submodular, and assumed to be normalized i.e., $f_{FL}(\varnothing) = 0$ and $f_{FL}(V) = 1$ (and the same for $c(\cdot)$). This normalization ensures the compatibility among the mixture components and the resulting mixture function $g_\lambda$ inherits these properties.

In Alg. 1, we present the offline KV cache summarization pseudo-code to compute a summary of the KV cache under a cardinality constraint $\tau_s$. This algorithm is used for the KV cache summarization in each self-attention head present in different decoder layers. In Line 3, we first compute the attention scores $a_n$ for all keys accumulated over different time steps and then use that to instantiate our feature-based function $c(\cdot)$. In our current setting, we only experiment with one feature function, meaning $|U| = 1$ in Eq. 3. The non-negative weight $m_u(k_i)$ in Eq. 3 for key $k_i$ is $a_n^i$, which represents the accumulated attention $k_i$ receives from the observed input queries. Using the key embeddings, we instantiate the facility location function $f_{FL}(\cdot)$ on a similarity matrix computed using pairwise cosine similarities followed by the ReLU transformation.

In Lines 4-7, we use the greedy algorithm (Nemhauser et al., 1978; Minoux, 1978; Mirzasoleiman et al., 2015) to perform a cardinality-constrained submodular maximization. Thanks to submodularity, the resultant set is within a factor of $(1 - 1/e)$ from the optimal summary (Nemhauser et al., 1978). This offline routine is suitable for KV cache summarization after the prefill stage of LLMs, particularly in serving systems where prompts are shared across user requests, and their KV embeddings are pre-computed and cached (Kwon et al., 2023; Gim et al., 2023).

In Alg. 2, we outline the KV cache summarization algorithm suitable for a streaming setting where we do not have prior knowledge of the complete sequence, restricting us from using existing offline summarization algorithms to obtain a global summary of the sequence. In light of this limitation, we propose to summarize the sequential data observed so far and produce an online summary of a fixed size that serves as its representative set. We remark that the computation at each step is at most $O(\tau_s^2)$ and which is fixed regardless of the length $n$ of the sequence. Further discussion is given in Appx. E. We use the same mixture function as Alg. 1 as our final scoring function.

In Lines 6-8, we keep caching the (key, value) pairs for different attention heads unconditionally until the summarization budget is exhausted. Once the KV cache is full, we utilize the key embeddings in $S_t$ and the accumulated attention score vector $a_t$ to update the submodular function components $f_{\text{FL}}(\cdot)$ and $c(\cdot)$ of the final mixture function $g_\lambda(\cdot)$. In Line 10, we create set $S_t'$ that includes the newest incoming $(k_t, v_t)$ pair and evaluate the conditional gain of keeping an item around in the context of the remaining summary set as shown in Line 11. The element with the least conditional gain is removed from $S_t'$ and the accumulated attention vector $a_t$ is modified in Lines 13-15 to include the attention score associated with the newest key when the discarded key is one of the keys in the previous summary set $S_{t-1}$.

The streaming summarization algorithm (Algo. 2) is suitable for memory-constrained settings where the KV cache for the entire context cannot be maintained in the GPU memory. Also, in multi-turn dialogue systems (Duan et al., 2023; Maharana et al., 2024), after a certain number of interactions, it can be challenging to keep track of the entire conversation context. In such scenarios, *BumbleBee* can enable modern LLM-based serving systems to maintain a representative yet important summary of past interactions while minimizing the KV cache-based memory utilization.

## 6 Experiments

### 6.1 Datasets & Tasks

The datasets studied in this paper are derived from three benchmarks: *lm-eval-harness* (Gao et al., 2023), *HELM* (Liang et al., 2022), and *LongBench* (Bai et al., 2023). Following Heavy-Hitters (Zhang et al., 2023), we select the following six few-shot datasets from lm-eval-harness: OpenbookQA (Mihaylov et al., 2018), COPA (Roemmele et al., 2011), RTE (Wang et al., 2018), MathQA (Amini et al., 2019), PiQA (Bisk et al., 2020), and Winogrande (Sakaguchi et al., 2021). From the HELM benchmark, we choose the single document summarization dataset XSUM (Narayan et al., 2018). From the LongBench benchmark (Bai et al., 2023), which is meant for evaluating the long-context understanding of LLMs, we select four tasks and their associated datasets: (1) Single document question answering: Qasper (Dasigi et al., 2021) and MultiFieldQA, (2) Multi-document question answering: HotpotQA (Yang et al., 2018) and 2WikiMultihopQA (Ho et al., 2020), (3) Summarization: QMSum (Zhong et al., 2021), and (4) Few-shot learning: TREC (Li & Roth, 2002).

All submodular functions and their optimizations are implemented with an upcoming to-be-open-sourced optimized C++-based software system called *Submarine* (Bilmes, 2024). All submodular computation, which is non-SIMD-style mixed integer-floating point, is done on the multi-threaded CPU side since that is a natural, cost-effective, efficient and thus still performant platform for such computation than vector-capable GPUs.

### 6.2 Evaluated Models

We use LLaMA 7B and 13B models (Touvron et al., 2023a) for tasks belonging to the lm-eval-harness benchmark. For the XSUM dataset, we use LLaMA2 7B and 13B models (Touvron et al., 2023b). On the LongBench selected datasets, we use the Llama-2-Chat 7B fine-tuned model and the LongChat-32k 7B model (Li et al., 2023).

### 6.3 Baselines & Methods

To show the effectiveness of *BumbleBee* for the context (KV cache) summarization task, we compare it to the following baselines: (1) **All**: we use the entire KV cache and do not perform any cache reduction (2) **Local**: only the most recent context x% tokens are maintained in the KV cache and the remaining old KV states are evicted (3) **Random + Local**: randomly selected tokens along with the most recent tokens are retained in the KV cache (4) **Attention sinks + Local**: the first four tokens known as attention sinks (Xiao et al., 2023) along with the most recent tokens are kept in the KV cache (5) **H2 + Local**: only the tokens that are most frequently attended to, referred to as Heavy Hitters (Zhang et al., 2023) are maintained in the KV cache along with the most recent tokens.

| Model | Methods | OpenBookQA | COPA | RTE | MathQA | PiQA | Winogrande |
|-------|---------|-----------|------|-----|--------|------|-----------|
| LLaMA-13B | **All** | 47.4 | 85 | 73.28 | 31.86 | 80.36 | 75.69 |
| | **Local** | 28.4 | 64 | 53.43 | 23.25 | 58.32 | 49.88 |
| | **Random + Local** | 27.6 | 58 | 54.63 | 21.76 | 54.13 | 50.64 |
| | **Attn Sinks + Local** | 44.4 | 80 | 67.51 | 29.78 | 79.22 | 70.48 |
| | **H2 + Local** | 44.2 | 83 | 64.98 | 29.71 | **79.49** | 70.32 |
| | **BumbleBee ♥** | **47.6** | **85** | **71.48** | **31.02** | 79.38 | 71.98 |
| | **BumbleBee ♦** | 46.6 | 83 | 67.15 | 30.82 | **79.49** | **73.01** |
| LLaMA-7B | **All** | 44.6 | 81 | 68.95 | 29.85 | 80.03 | 71.51 |
| | **Local** | 28.4 | 56 | 50.90 | 23.02 | 58.27 | 51.38 |
| | **Random + Local** | 28.0 | 63 | 51.26 | 21.76 | 53.94 | 49.30 |
| | **Attn Sinks + Local** | 41.6 | **82** | 58.12 | 27.40 | 78.07 | 67.80 |
| | **H2 + Local** | 41.4 | 78 | 63.54 | 27.50 | 77.31 | 65.82 |
| | **BumbleBee ♥** | 43.2 | 79 | **68.95** | 27.74 | 78.24 | **68.75** |
| | **BumbleBee ♦** | 43.2 | 79 | 63.90 | **28.51** | **78.56** | 68.19 |

Table 1: Results on the few-shot tasks from the lm-eval-harness benchmark using LLaMA 7B and 13B (Touvron et al., 2023a). In the above methods (except All), we do a $10\times$ context reduction, so our KV cache summarization budget is $0.1\times$ the input sequence length.

| Model | Method | Qasper | MultiFieldQA-en | HotpotQA | 2WikiMQA | QMSum | TREC |
|-------|--------|--------|-----------------|----------|----------|-------|------|
| LLaMA-7B-chat 4k | **All\*** | 19.20 | 36.80 | 25.40 | 32.80 | 20.80 | 61.5 |
| | **All** (self) | 21.60 | 36.76 | 27.55 | 31.58 | 20.78 | 64.0 |
| | **Attn Sinks + Local** | 14.74 | 22.93 | 22.08 | 29.73 | 19.25 | 56.0 |
| | **H2** (20%) | **19.82** | 26.60 | 26.28 | 25.69 | **21.45** | 60.0 |
| | **BumbleBee** (20%) ♥ | 19.37 | 27.73 | 26.14 | 27.67 | 20.68 | **61.5** |
| | **BumbleBee** (20%) ♦ | 19.59 | **28.60** | **28.99** | **30.19** | 21.05 | 59.0 |
| LongChat-7B 32k | **H2** (SW, 20%) | 21.64 | 30.72 | 14.07 | 15.10 | 18.11 | 40.5 |
| | **BumbleBee** (SW, 20%) ♦ | **23.27** | **33.16** | **22.52** | **17.58** | **20.27** | **44.5** |

Table 2: Results on six datasets from the LongBench benchmark using *Llama2-7B-chat-4k* (Touvron et al., 2023b) and *LongChat-v1.5-7B-32k* (Li et al., 2023). * indicates that the reported numbers are sourced directly from Bai et al. (2023).

In the case of *BumbleBee*, we test two concave function choices for Eq. (3): (♥) log-based: $\phi(x) = \log(1 + x)$ and (♦) power-based: $\phi(x) = g^{-1}(x)$ where $g(y) = \alpha y^{1/\alpha} + \beta y$. Herein, we set $\beta$ and $\alpha$ as 1 and 0.04 respectively to ensure that the function saturation curve is compatible with the facility location function, the first submodular function component of $g_\lambda(\cdot)$ in Eq. 4, thus ensuring that neither component from Eqn (3) or Eqn (2) ever always dominates the other in the optimization. Selecting the form and curvature of the concave function in the submodular mixture is a hyperparameter that is just as important for good BumbleBee performance as is the mixture parameter $\lambda$ in Eqn. (4). Hyperparameters were tuned on a development and tested on a test set (see Appx. D). The extreme cases ($\lambda \in \{0, 1\}$), corresponding to using only one component from Eqns (3) and (2), yield suboptimal performance.

## 6.4 Results

**LM-eval-harness Tasks**: In Table 1, we compare *BumbleBee* to existing KV cache reduction methods on five-shot learning tasks from the lm-eval-harness benchmark (Gao et al., 2023). Across both LLaMA (Touvron et al., 2023a) variants we used for inference, *BumbleBee* consistently outperforms other baselines in terms of accuracy, showing a performance comparable to the best-case no-compute-constraint setting when *All* of the KV cache is used without any summarization/reduction.

**LongBench Tasks**: In Table 2, we report the results on six different datasets from Long-Bench (Bai et al., 2023). For all datasets excluding QMSum and TREC, we use F1 score for evaluation. For QMSum and TREC, we use Rouge-L and accuracy respectively as the evaluation metric. When using the *LLaMA-7B-chat-4k* model pre-trained with a 4k context window, we truncate the middle part of the input contexts if their length exceeds 4k as suggested in Bai et al. (2023). As can be seen in Table 2, the offline version of *BumbleBee* outperforms other context reduction methods such as *H2* on four datasets even when the summarization budget is 20% of the entire context size.

However, when using the *LongChat-7B-32k* which was trained further to generalize to longer sequences, we adopt a Sliding Window (SW) strategy to process smaller chunks/segments of longer contexts, aggregate their KV embeddings, and the attention scores accumulated for keys in different chunks. This is done to process the entire context/sequence in a memory-manageable way while avoiding context truncation. This strategy however is sub-optimal as tokens in faraway segments cannot attend to tokens at the beginning of the original sequence. Despite this, we observe that *BumbleBee* outperforms the state-of-the-art KV cache reduction method *H2* by strong margins (1.6%-8.5% in absolute terms). In Appx. C, we present qualitative results that illustrate how BumbleBee effectively retains a representative subset of keys within the reduced KV cache.

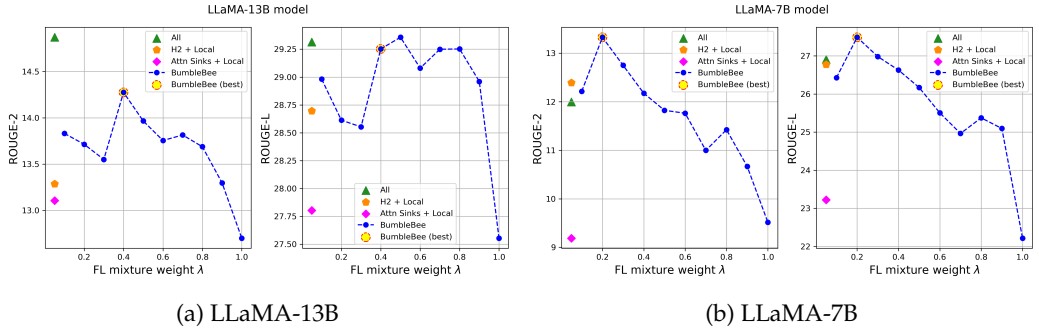

(a) LLaMA-13B           (b) LLaMA-7B

Figure 3: ROUGE-based results on XSUM dataset, a few-shot summarization task from the HELM benchmark (Liang et al., 2022) using two different LLaMA models (Touvron et al., 2023b). To reduce the pressure on the context window across all decoder layers, we perform a 5x KV cache reduction for each of the above methods except *All*.

**XSUM Summarization Task**: In Fig. 3, we compare different context reduction methods to the full cache setting. We use LLaMA2 models (Touvron et al., 2023b) to assess the downstream summarization performance in a 3-shot learning setting. *BumbleBee* outperforms other SOTA cache reduction techniques and even performs better than the *All* cache setting when using LLaMA-7B.

**Sensitivity analysis of mixture weight** $\lambda$: We use the XSUM dataset to show the overall performance sensitivity of *BumbleBee* to the convex mixture weight hyperparameter $\lambda$ in Eq. 4. In the case of LLaMA-13B model, $\lambda \in [0.4, 0.8]$ performs comparably showing that a representative subset of the KV cache is preferred for the downstream task. In LLaMA-7B, we see that for $\lambda > 0.2$, the performance starts to drop as we increase the mixture weight $\lambda$ corresponding to the Facility location function $f(\cdot)$. However, the first submodular component with its relative weight of 0.2 still outperforms the *H2 + Local* method showing that both representativeness and relevance of the selected cache subset are desirable to maintain a performance comparable to the entire cache setting.

# 7   Conclusions & Future Work

For future work, we plan to evaluate *BumbleBee* on reasoning (Sawada et al., 2023) and problem-solving datasets (Cobbe et al., 2021) and investigate the impact of hyperparameter tuning and alternative function mixtures in our formulation. Additionally, we plan on modifying sliding-window attention with a submodular formulation to better capture long-term dependencies. More generally, regarding human episodic memory and selective attention (Chun & Turk-Browne, 2007; Uncapher & Rugg, 2009), it will be interesting to consider and study the dynamic "bounds", "limits", and "finite capacity" of episodic human memory such as (Brehmer et al., 2004; Dings & McCarroll, 2022). With BumbleBee, this corresponds to the size of the historical summary that is retained. It may be that in human memory, this bound grows slowly over time. A corresponding BumbleBee variant could be produced where the submodular summary size grows very slowly over time as well (e.g., $\log(1 + \log(1 + \log(1 + \ldots n))))$. The attention layers should be able to support this.

**Acknowledgments:** We thank Harshil Dadlani for running synthetic experiments. This material is based upon work supported by the National Science Foundation under Grant No. IIS-2106937 and is supported in part by funds from federal agency and industry partners as specified in the Resilient & Intelligent NextG Systems (RINGS) program Grant No. IIS-2148367.

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

## A  Supplementary Notes to the Introduction

We remark that certain deep neural networks such as Recurrent Neural Networks (RNN) (Rumelhart et al., 1986) and their variants such as Long-Short Term Memory networks (LSTM) (Hochreiter & Schmidhuber, 1997) and Gated Recurrent Units (GRU) (Cho et al., 2014) have been widely used for sequential data (Connor et al., 1994; Mikolov et al., 2010; Sundermeyer et al., 2012). Despite their strong performance, RNNs suffer from vanishing and exploding gradient problems (Hochreiter et al., 2001) as well as an inherent Markov property similar to dynamic Bayesian networks, where the future and past are separated by (or independent given) the present, thereby limiting their ability to capture extremely long-term dependencies without extremely large state spaces (Bilmes, 2006). Thanks to its transformer underpinnings, a BumbleBee model does not have any such Markov property even in the "infinite-context" case.

# B    Experiments on Synthetic Data

We are given a dataset having inputs defined as $x = \{x_1, x_2, \ldots, x_n\}$ where $x_t \in \mathbf{R}^{d_{in}}$ denotes the t-th input token and $d_{in}$ is the input feature dimensionality. Similarly, we have the outputs defined as $y = \{y_1, y_2, \ldots, y_n\}$ where $y_t \in \mathbf{R}^{d_{out}}$ and $d_{out}$ is the dimension of the output embedding. We hypothesize that *BumbleBee* can achieve significant improvements in the task involving estimating $y_t \in \mathbf{y}$ over existing methods across different dimensions if $y_t$ is a function of both of the following:

(a) The local context window, *local$_t$*, a sequence $\{x_{t-h}, x_{t-(h-1)}, \ldots, x_{t-1}\} \subseteq x$ where $h$ is the local context window length.

(b) The summary sequence, *summ$_t$*, comprising $k$ relevant vectors chosen from $\{x_1, x_2, \ldots, x_{t-h-1}\} \subseteq x$. Specifically, let $S_t \subseteq \{1, 2, \ldots, (t - h - 1)\}$ and say $S_t = (s_1, s_2, \ldots, s_k)$ is ordered by time (or position) in the original sequence so that $s_1 < s_2 < \cdots < s_k$. Then we have that $summ_t = (x_{s_1}, x_{s_2}, \ldots, x_{s_k})$.

Also, assume in this discussion that $t \geq h + k$ to avoid any negative indices.

To understand the context where we expect *BumbleBee* to function well, we define an (undesirable) conditional independence property as follows:

$$y_t \perp\!\!\!\perp summ_t | local_t \tag{5}$$

This property states that the predicted variable $y_t$ is independent of the summary given the local context — this means that the local context is sufficient to predict $y_t$ and once we have this local context, the summary is not needed. This property is precisely what we must **not** have in the data for *BumbleBee* to function properly. That is, this property should not hold for any of the estimation tasks under consideration.

In terms of conditional mutual information, the property that we would like to have is that $I(y_t; summ_t | local_t) > 0$, meaning that even if we have the local context, the summary is still informative of $y_t$. This property being true means *BumbleBee* should be effective.

To demonstrate the effectiveness of the *BumbleBee* on such tasks, we define an ideal experimental setup as follows:

We obtain $x \sim \mathcal{N}(0, 1)$, for each $t \in \{(k + h), (k + h + 1), \ldots, N\}$, we sample $local_t \in x$ using a sliding window mechanism and $summ_t \in x$ uniformly at random based on the window size and summary budget size defined previously. We define a generative process to obtain $y$ as follows:

$$y = f_{\text{ground-truth}}(summ, local)$$

$f_{\text{ground-truth}}$ is chosen in such a way that the summary and local context tokens are both equally important to $y$. This is done by obtaining,

$$y^{local} = f_{\text{ground-truth}}(local)$$

$$y^{summ} = f_{\text{ground-truth}}(summ)$$

where the above model $f_{\text{ground-truth}}$ is achieved via standard transformer masking/padding (setting the corresponding attention values in the matrix to zero), and then comparing their Mean Squared Error (MSE) scores with respect to $y$ to be comparable as shown in Table 3. Note that, we are not training the ground-truth model, but rather initializing it with a seed for which the predictions made using (a) only the local tokens and (b) only using the summary tokens receive similar MSE values. In terms of the architecture, $f_{\text{ground-truth}}$ is a single-layer encoder-decoder transformer model with a linear projection layer applied to the outputs of the decoder. Here, $h = 32$, $k = 32$, $d_{in} = 16$, $d_{out} = 1$, $n = 80{,}000$.

To demonstrate our initial hypothesis, we now perform training and obtain three different learnt models, as follows:

| Model | MSE (Mean Squared Error) |
|---|---|
| $f_{\text{ground-truth}}(summ)$ | $8.2689 \times 10^{-4}$ |
| $f_{\text{ground-truth}}(local)$ | $8.3301 \times 10^{-4}$ |

Table 3: Performance comparison of the generated ground-truth model on the entire dataset when only the summaries are used and when only the local context is used

a. $f_{learnt}^{summ,local}$ - trained using both summaries and local context tokens.

b. $f_{learnt}^{local}$ - trained using only the local context tokens.

c. $f_{learnt}^{summ}$ - trained using only the summary tokens.

For the entire training setting, unless mentioned otherwise, we perform an 80:10:10 train, validation, and test split respectively (random seed initialization same across all three settings), and train for 50 epochs. The results are evaluated on the held-out test set. From Table 4 and Fig. 4, it is clear that $f_{learnt}^{summ,local}$ performs significantly better than $f_{learnt}^{local}$ and $f_{learnt}^{summ}$, exhibiting the importance of both summary and local context input tokens for the final predictions. In Fig. 5, we provide qualitative results showing the predicted values using different learned models.

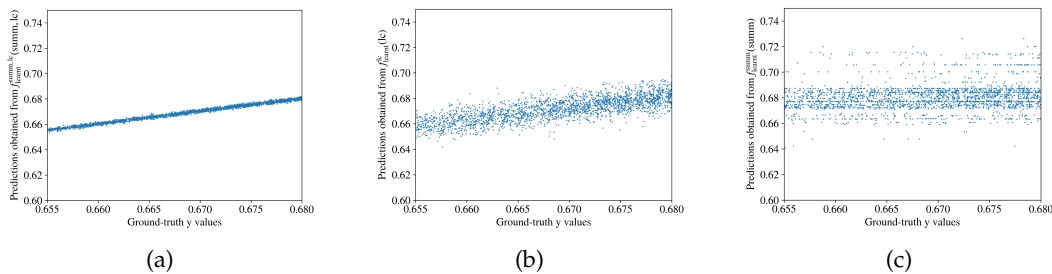

| (a) | (b) | (c) |
|---|---|---|

Figure 4: Performance comparison on a subset of the held-out set for different models trained using (a) summaries and local context; (b) only local context; (c) only summaries. Note that the plot in (c) looks granular in the vertical dimension since the summaries are not changing at every time point, i.e., time is segmented into regions where the summary is fixed, meaning $S_t = S_{t-1}$ for certain regions.

We observe the stripe pattern in Fig. 4c because the input summary tokens remain constant across multiple episodic periods due to the design of the data sampling process.

| Model | MSE (Mean Squared Error) | MAE (Mean Absolute Error) |
|---|---|---|
| $f_{learnt}^{summ,local}$ | $9.14 \times 10^{-7}$ | $7.7279 \times 10^{-4}$ |
| $f_{learnt}^{local}$ | $4.21 \times 10^{-5}$ | $5.09 \times 10^{-3}$ |
| $f_{learnt}^{summ}$ | $4.83 \times 10^{-4}$ | $1.685 \times 10^{-2}$ |

Table 4: Model performance on held-out test set

We further demonstrate that using a large local context does not necessarily yield comparable results using the same training setting. To achieve this, we simply increase the size of the local context window by 10 times the original to obtain, $f_{learnt}^{local \times 10}$. We compare the performance of these models on metrics like MAC (Multiply and Accumulate), runtime memory, and MSE. The MAC and runtime memory metrics are computed for a single training example in a mini-batch.

Table 5 clearly shows the effectiveness of having a mechanism to summarize the historical

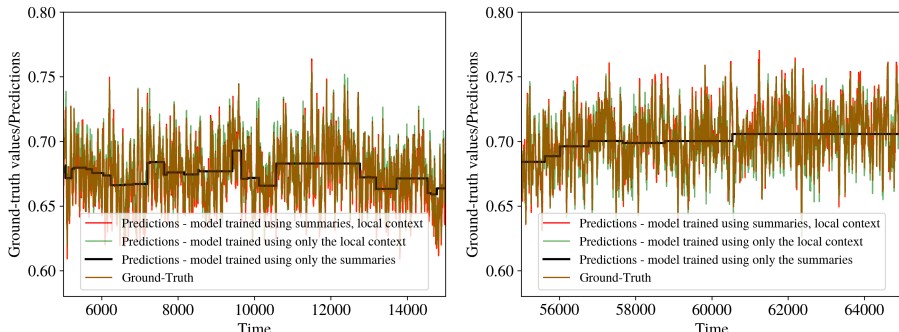

Figure 5: Plot for error comparison with respect to ground-truth data of different trained models for two different time windows

context over naively fitting longer inputs in the transformer's context in terms of metrics such as runtime memory used, MSE, etc.

| Model | MSE (Mean Squared Error) | MAC | Runtime Memory |
|---|---|---|---|
| $f_{learnt}^{summ,local}$ | $9.14 \times 10^{-7}$ | $6.33 \times 10^5$ | 3.43 MB |
| $f_{learnt}^{local \times 10}$ | $3.43 \times 10^{-4}$ | $3.12 \times 10^6$ | 5.98 MB |

Table 5: Performance comparison of a model trained using a naive large local context to a *BumbleBee* model

## C   Detailed Analysis of Results

For one of the **lm-eval-harness** (Gao et al., 2023) tasks namely COPA, we analyze how the KV cache across different self-attention heads gets updated as new queries are processed in a streaming setting. In Figures 6 & 7, we show such visualization for two samples selected from the test set of COPA (Roemmele et al., 2011). Since each incoming query attends to its key vector, the offset diagonal line simply indicates that property. Overall, we see that *H2* is heavily biased towards selecting the initial set of keys (close to position 0) and maintaining them in the KV cache. This pattern holds across most of the attention heads. However, *BumbleBee* summaries appear more time-diverse across the majority of heads showing that our framework is capable of balancing both representativeness and relevance in the final summary of the KV cache.

Next, we visualize how good the subsets selected by Heavy Hitters (H2) and *BumbleBee* are on one of the samples chosen from the 2WikiMultihopQA (Ho et al., 2020) dataset from the LongBench task. We visualize the t-SNE embeddings of the keys across randomly selected self-attention heads in certain layers along with the selected subset of the KV cache. Fig. 8 shows that *BumbleBee* can select a representative subset of keys from the entire KV cache when using the offline greedy algorithm, thus explaining its strong performance on various LongBench datasets.

Table 6 compares the decoding speeds (in ms/token) for two different context reduction ratios, demonstrating faster decoding speed as the context reduction ratio becomes higher.

## D   Reproducibility

We implement *BumbleBee* in PyTorch. To update the KV cache in the streaming setting, we modify the *LlamaAttention* class from the huggingface library. For the submodular optimization, we use an internal highly optimized submodular toolkit, and we plan to

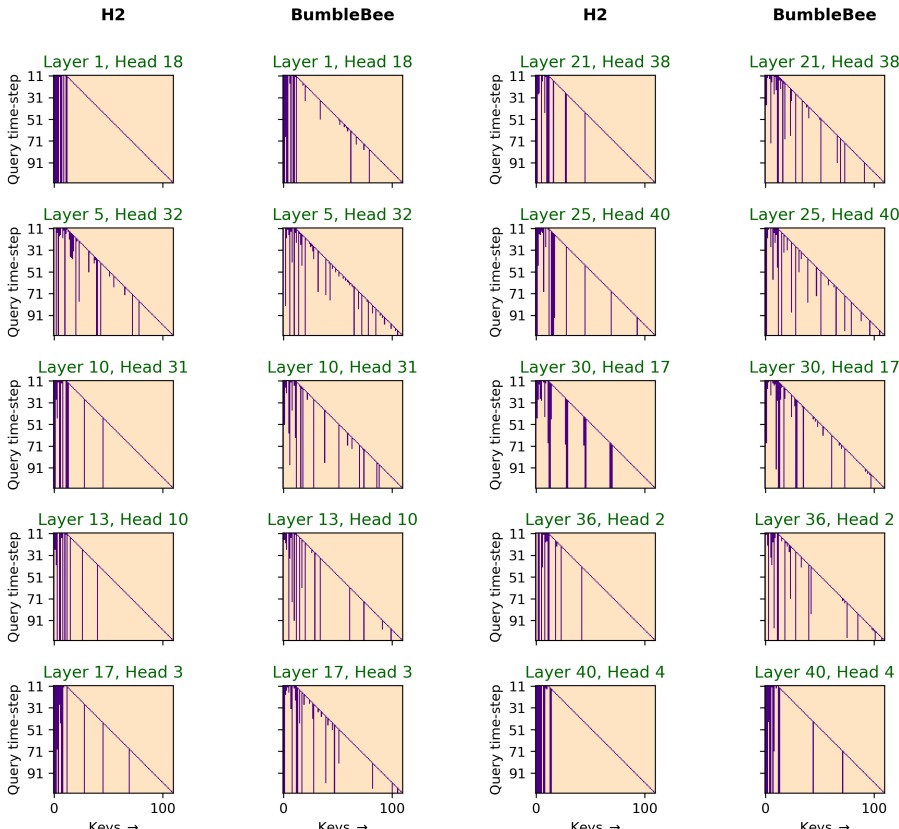

Figure 6: Test sample 1: Visualization of the keys selected at different time-steps (y-axis) when using *H2* and *BumbleBee* for online KV cache reduction and summarization respectively. The x-axis represents the keys, and the y-axis represents the queries. We show the evolution of the selected KV cache (marked by purple points) as we process an incoming query across different attention heads and determine how to update the KV cache under the budget constraint.

| Context reduction ratio | Original Context Length | |
|:---:|:---:|:---:|
| | **16k** | **100k** |
| **1:1** | $59.30 \pm 0.39$ | OOM |
| **5:1** | $47.49 \pm 4.16$ | $71.50 \pm 0.10$ |
| **10:1** | $39.74 \pm 1.31$ | $48.16 \pm 0.09$ |

Table 6: Decoding speed (in ms/token) for two KV cache reduction ratios (5:1 and 10:1) and the baseline KV cache method using the entire context (1:1) across all heads. All experiments are performed on an A100 80GB GPU using the LongChat-7B-32k with a batch size of 1.

open-source the integrated codebase shortly.

On the LongBench dataset, we first compute the key embeddings for all the context tokens. This is performed for each self-attention head in each decoder layer. We then use stochastic greedy algorithm (Mirzasoleiman et al., 2015) with $\epsilon = 1e-5$ to compute the offline KV cache summaries for each of $h \times l$ heads in a parallel fashion. Here, $h$ denotes the number of attention heads present in one decoder layer, and $l$ denotes the number of decoder layers.

For similarity matrix computation, we use ReLU truncated cosine similarity to ensure pairwise similarities $sim(i, j) \geq 0$. We have also experimented with other cosine similarity-based metrics such as $1 + \cos(i, j)$ and $|\cos(i, j)|$ but find that $ReLU(\cos(i, j))$ works the best.

Mixture weight $\lambda$: on the tasks from the lm-eval-harness benchmark (Gao et al., 2023),

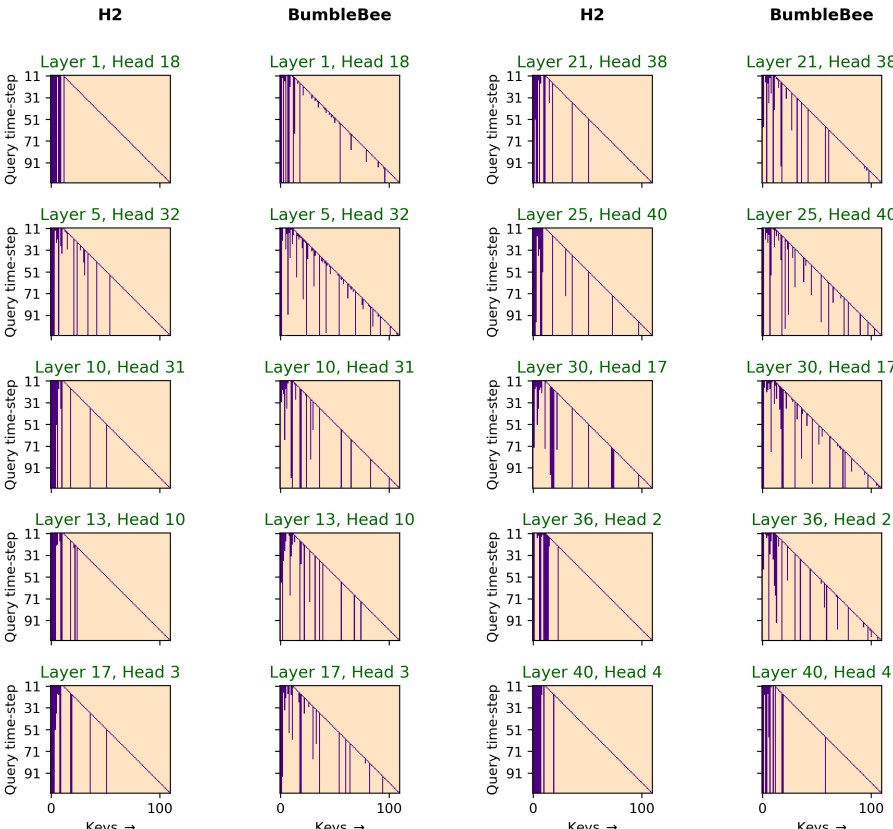

Figure 7: Test sample 2: Visualization of the keys selected at different time-steps (y-axis) when using *H2* and *BumbleBee* for online KV cache reduction and summarization respectively. The x-axis represents the keys, and the y-axis represents the queries. We show the evolution of the selected KV cache (marked by purple points) as we process an incoming query across different attention heads and determine how to update the KV cache under the budget constraint.

$\lambda \in \{0.2, 0.3\}$ perform the best across both evaluated models, i.e., LLaMA 7B and 13B. However, we did not perform a more fine-grained search/tuning for $\lambda$ in this range. On LongBench (Bai et al., 2023), we report the results for $\lambda = 0.3$ across all six datasets. For the XSUM (Narayan et al., 2018) few-shot summarization task, we use a held-out validation set to tune the mixture weight and set $\lambda = 0.2$ when using LLaMA2-7B and $\lambda = 0.4$ for LLaMA-13B.

Compute: we use an NVIDIA-A100 GPU to perform our inference-based experiments and a multi-threaded CPU for all submodular computation using Submarine (Bilmes, 2024). For the experiments involving training on synthetic data in Sec. B, we use an NVIDIA RTX 2080.

## E   Complexity Analysis

**Compute costs**: In BumbleBee (Alg. 2), we maintain a global summary of size $\tau_s$. So, constructing the pairwise similarity matrix for $f_{\text{FL}}$ has a time complexity of $\mathcal{O}(\tau_s^2 \times d)$. Identifying the item with the least conditional gain (Line 11 of Alg. 2) requires $\mathcal{O}(\tau_s^2)$. However, if we cache the similarity matrix, we only need to compute the similarity of the incoming item to the others in the summary, resulting in an overall complexity of $\mathcal{O}(\tau_s \times d + \tau_s^2)$ for each new token.

**Memory Costs**: Caching the similarity matrix incurs $\mathcal{O}(\tau_s^2)$ memory costs.

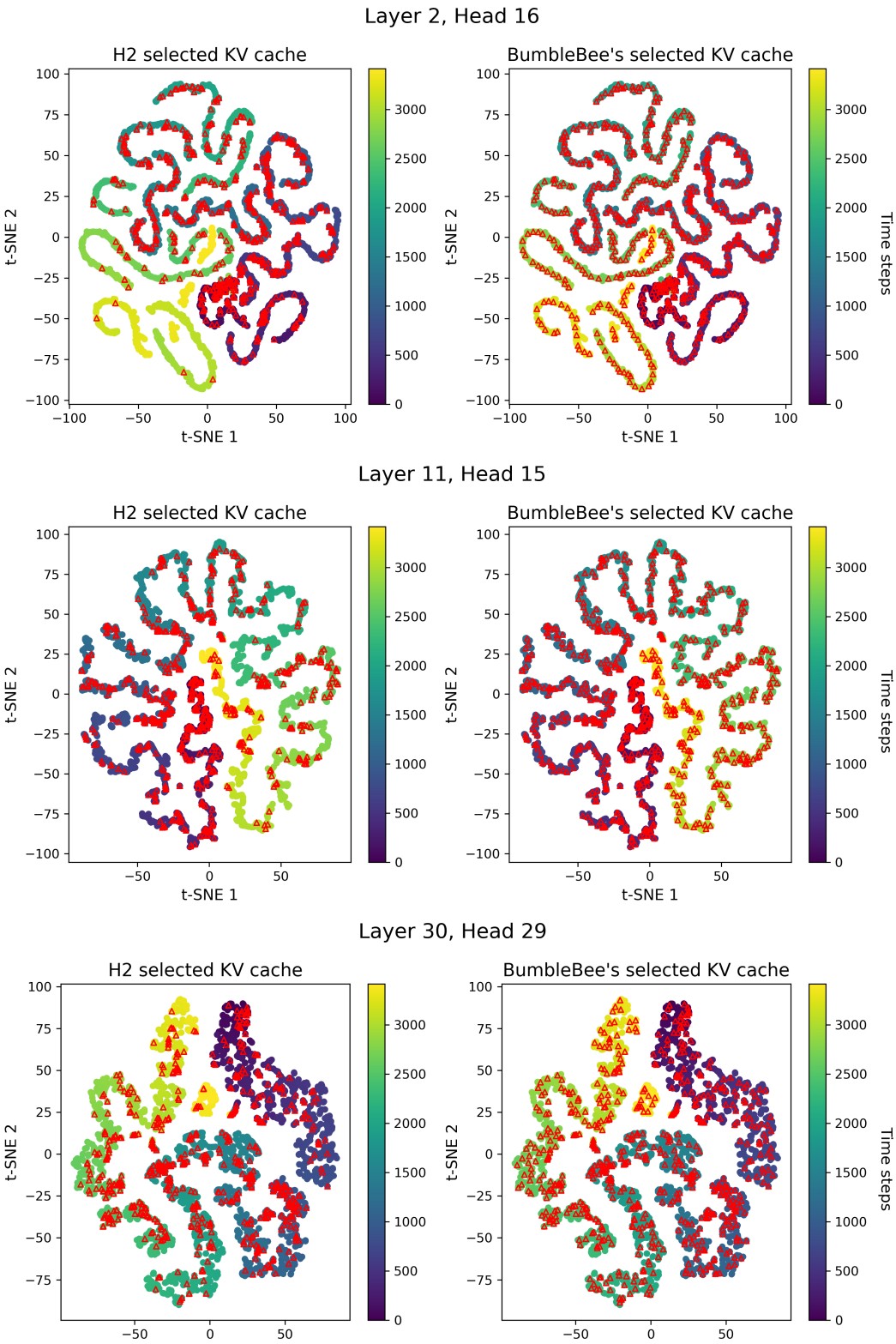

Figure 8: t-SNE visualization of the key embeddings for different attention heads in the LLaMA-7B-chat-4k model. The keys selected by *H2* and *BumbleBee* as a part of the final KV cache summary are marked by △.

