# OpenReview forum: "BumbleBee: Dynamic KV-Cache Streaming Submodular Summarization for Infinite-Context Transformers"
_colmweb.org/COLM/2024/Conference — COLM_

### Official Review · Reviewer_zX3y · 2024-05-09

**Rating:** 7
**Confidence:** 4
**Ethics Flag:** 1

**Summary:**

This paper proposes BumbleBee, a KV cache summarization method. KV cache is a technique to speed up the decoding procedure by caching key-value pairs computed in the previous steps. KV cache can avoid redundant computations, however, the number of cache entries grows linearly with the number of tokens and can be huge for recent LLMs dealing with long sequences. The proposed method reduces the amount of KV cache by selecting appropriate subsets of key-value pairs. The paper formulates the subset selection problem as an online submodular function maximization problem under cardinality constraints and proposes a simple greedy procedure to select appropriate subsets of key-value pairs for each generation step.

The paper combines BumbleBee with multiple LLMs and evaluates the performance in multiple tasks. Experimental results show that the proposed method outperforms baseline KV cache reduction methods.

**Questions To Authors:**

1. Is there any theoretical guarantee for the online submodular maximization algorithm of Algorithm 2?

2. How do we compute $m_u(\cdot)$?

3. Tab. 5 and Fig. 5 seem not to be referenced from the main body of the paper.

**Reasons To Accept:**

1. The idea of formulating a KV cache reduction problem as a subset selection problem with a submodular objective function is interesting. I agree that employing submodular objective functions is a reasonable choice for the KV-cache reduction.

2. The proposed method is simple, and we do not need additional complex procedures like fine-tuning.

3. Experimental results show that the proposed method outperforms baseline KV-cache reduction methods.

4. The paper is well-organized and easy to follow.

**Reasons To Reject:**

1. The paper does not report the decoding speed of the proposed method. Since the KV cache is introduced to speed up the decoding procedure, I think it is important to show that BumbleBee is faster than the baseline KV cache method.

2. The paper reports experimental results for fixed budget sizes. The paper employs different budget sizes for LM-eval-harness tasks and LongBench tasks, but the reason why doing so is not explained. Budget size is an important parameter that can directly trade the performance with the GPU memory and running time. Therefore, it would be nice to show that the proposed method outperforms the baseline methods if we change the budget size.

---

> ### Author Rebuttal · Authors · 2024-05-31
>
> Thank you for your positive feedback and questions!
>
> **RTR1**: Please see https://imgur.com/a/1Bo8d8v
>
>
> **RTR2**: For LongBench datasets with offline BumbleBee, only tokens from the test input are kept in local window. The associated context (passages, transcripts) is summarized to 20% of its length at each attention head, ensuring the summarized KV cache generalizes across different test inputs. On lm-eval-harness, we provide results for the smallest summarization budget where performance loss is minimal compared to the no KV cache reduction setting.
>
>
> **Q1**: The current algorithm does not have a theoretical guarantee w.r.t. the non-linear transformer. The reason for this is multi-faceted: (1) Algorithms like Sieve-Streaming (Badanidiyuru et al. 2014), Sieve-Streaming++ (Kazemi et al. 2019), etc. assess the final summary only after the entire data stream has been processed, which is not conducive to the needs of real-time LLM decoding where continuously optimal summaries are essential (2) Integrating these into transformers further complicates theoretical proofs concerning performance or convergences due to non-linear nature of transformers (3) Unlike simpler feature-based functions, analyzing facility location functions in a streaming setting is not viable as the ground set changes as the stream is updated.
>
> Our summary update algorithm (Line 11, Alg 2) is, however, exact at each step - the reason, given the knowledge of the sequence observed so far at time t and running summary $S_{t-1}$ of fixed size $\tau_s$, it discards the single item with the least marginal gain ensuring that the function valuation of resultant summary is maximum amongst all $(\tau_s+1)$ candidate summaries.
>
>
> **Q2**: $m_u(v)$ represents the accumulated attention the key $v$ receives from observed input queries. The subscript $u$ represents different features that one might want to model to capture context relevance, for example, in grouped-query attention (GQA), each $u$ could correspond to a specific query head meaning $|U|$ would be the size of each query group.
>
>
> **Q3**: Thank you for pointing this out. We will fix this in our updated manuscript. Table 5 corresponds to the synthetic data experiment where we naively increase the size of the local context window by 10 times the original and compare it to the setting where the model uses the summarized context. Fig. 5 supports the findings of Table 4 by qualitatively showing the predicted values using different learned models.

---

> > ### Comment · Reviewer_zX3y · 2024-06-03
> >
> > Thank you for the response! I will raise my score since the authors address my concerns appropriately.

---

> > > ### Author Response · Authors · 2024-06-06
> > > **Thank you**
> > >
> > > We thank the reviewer for carefully reviewing our rebuttal and taking the time to revise their assessment. We are glad that our response addressed the main concerns, and we greatly appreciate your detailed feedback and suggestions throughout the review process.

---

### Official Review · Reviewer_aHg4 · 2024-05-10

**Rating:** 6
**Confidence:** 4
**Ethics Flag:** 1

**Summary:**

This paper proposes a submodular optimization framework called BumbleBee to compress the KQ cache in a long context. They aim to maintain the most representative KQ embeddings in history evaluated based on similarity and representativeness. The KV cache is iteratively updated at each decoding step. They validate BumbleBee's effectiveness on extensive tasks with impressive cache budget reductions.

**Questions To Authors:**

1. What is the value of \tau_s and \tau_l? Which is more important, global summary or local history?

2. What is the cost of BumbleBee memory and time? Do you save embeddings similarity (as shown in Equation 2) and the past attention weights used in equation (3) in the cache?

3. What are the results of "ALL" for LongChat-7b-32k in Table 2?

4. Why are the attention map values in Figure 2 negative? The colors in this figure are all red and orange, making it difficult to observe the region that is most attended.

**Reasons To Accept:**

1. A novel method to compress the KV cache.
2. Large reduction in KV cache memory.

**Reasons To Reject:**

1. Updating the summary KV cache summarization is time consuming. Step 11 in Algorithm 2 and step 5 in Algorithm 1 need to calculate the loss of removing each KV cache embedding. I wonder if this process can be optimized or computed in parallel.

---

> ### Author Rebuttal · Authors · 2024-05-31
>
> Thank you for the review and insightful comments! Please see our responses below:
>
> **RTR1**: This step to determine the key-value item to discard has a time complexity of $\mathcal{O}(\tau_s^2)$ (including FL evaluations). This process, however, can run concurrently with attention output computations in the subsequent decoder layer. Also, the summary updates for different self-attention heads within a layer are performed in parallel. Hence, the overall method is scalable. We will clarify this in the next version.
>
>
> **Q1**: On lm-eval-harness and XSUM datasets, both $\tau_l$ and $\tau_s$ are set to 10% of the input sequence's length. For LongBench datasets with offline BumbleBee, only tokens from the test query/input are kept in the local window. The associated context (e.g., passages, transcripts) is summarized to 20% of its length at each attention head, ensuring the summarized KV cache generalizes across different test inputs. The ratio between  $\tau_l$ and $\tau_s$ should be task-dependent, for example, for context summarization tasks such as XSUM, a larger portion of the KV cache budget should be dedicated to the summary to cover diverse and representative aspects across the provided context (https://imgur.com/a/YWZ2aNK).
>
>
> **Q2**: In our current setup, we cache both accumulated attention scores and similarity matrix. Also, the per-layer summary update is overlapped with the attention output computations in the subsequent decoder layer.
>
> *Memory costs*: $\mathcal{O}(\tau_s)$ for each head when computing similarity matrix on-the-fly. $\mathcal{O}(\tau_s^2)$ when caching the similarity matrix.
> *Compute costs*: $\mathcal{O}(\tau_s^2 \times d + \tau_s^2)$ when similarity matrix is computed real-time. Here, $d$ is the key-embedding dimensionality and the 2nd term corresponds to the summary update costs (Line 11 of Alg. 2) . We cache the similarity matrix and thus only need to compute the similarity of the incoming item to others in the summary. In this case, therefore, it costs overall $\mathcal{O}(\tau_s \times d + \tau_s^2)$ for each new token.
>
>
> **Q3**: We do not report the results for ALL due to encountering OOM error on our GPU (A100 80 GB) when using a longer context (>32k tokens).
>
>
> **Q4**: In Fig. 2, we plot the logarithm of the attention scores as mentioned in the paper. For better visualization, we will add another figure that uses histogram equalization to improve contrast (similar to https://imgur.com/a/4bph47f).

---

> > ### Author Response · Authors · 2024-06-06
> > **Thank you**
> >
> > Thank you sincerely for your detailed and helpful review! We hope our responses have effectively addressed your concerns and would greatly appreciate the opportunity to further integrate your suggestions to enhance our work. Please let us know if you have any additional questions or comments.

---

### Official Review · Reviewer_wh5U · 2024-05-11

**Rating:** 7
**Confidence:** 4
**Ethics Flag:** 1

**Summary:**

This paper presents a technique to compress the KV cache in LLMs using a mixture of submodular functions to select a diverse and representative set of keys from the context. Experiments on a series of well-studied LLM benchmark datasets indicate that this technique typically outperforms prior work and is occasionally competitive with full-context models.

**Questions To Authors:**

* Is $m_u$ in eq. 3 defined as the attention weight for each key?

* Modern large-scale LLMs typically use multi-query or group-query attention. How does this approach need to be modified—if at all—in order to accommodate these techniques? For instance, would attention weights be accumulated separately for each query index?

* There are no results for BumbleBee 💙 in the LongChat-7B 32k experiments, and it is not clear which flavor of BumbleBee is used for Figs. 3, 6-8.

**Reasons To Accept:**

* The paper is well written and the proposed approach is clearly motivated.

* Submodular formulations are an intuitive choice to encourage diversity in the compressed KV cache.

* Experiments on a variety of benchmarks and models suggest the performance of this approach is fairly consistent.

**Reasons To Reject:**

* The computation of $g_\lambda(\cdot)$ appears fairly expensive, especially the FL function (eq. 2) in the streaming setting (algorithm 2), which seems to require $O(n^2 \tau_s d)$ time when computed naively. This may be a reasonable tradeoff for the memory savings, but the paper does not include an evaluation on the impact on latency. A highly-optimized toolkit for computing such functions is briefly mentioned in Appendix C but more information on runtime costs would be helpful.

* The relevance of the particular submodular functions chosen here to the task at hand is not conveyed clearly. It appears the feature-based function (eq. 3) may be computing the sum of attention scores for each key—similar to H2—but this is not explicitly described.

* The notation is overloaded and confusing. $V$ is used to represent both a grounding set and the set of value embeddings, even though the actual grounding set is based on keys rather than values. Similarly, $f$ is used to represent both a generic submodular function as well as the specific FL function (eq. 2). Accumulated attention scores in Algorithm 2 are denoted by $a_t$, thereby not using the set notation or indicating that they are specific to each key.

* The evaluation on the lm-eval-harness benchmark is based on the 15-month-old Llama models, and it is unclear whether the results would generalize to newer, stronger models that may use context more effectively.

* Although a number of papers on KV cache compression are cited, the evaluation only compares against H2 and Attention Sinks as baselines from prior work.

---

> ### Author Rebuttal · Authors · 2024-05-31
>
> Thank you for your detailed feedback!
>
> **RTR1**: In BumbleBee, we maintain a running summary of fixed size $\tau_s$, so our runtime complexity will not be a function of sequence length n. For FL function, as each key-value is observed, we compute its similarity to items in the KV cache summary incurring $\tau_s d$ cost. Computing $g_\lambda$ for each item in line 11 of Alg. 2 takes $\mathcal{O}(\tau_s^2)$ including FL evaluations. Multiple functions for different heads can be evaluated in parallel (as in our implementation).
>
>
> **RTR2**: H2 is a specific case of our formulation with $\lambda$ set to 0 and $\phi_u$ as an identity function. Our submodular objective uses the FL function to capture diversity in key embedding space and a concave function to transform modular attention scores into a strictly submodular objective. For larger models using grouped-query attention GQA, the concave function $\phi_u$ in Eq. 3 can be chosen selectively to control the rate of diminishing returns for different query heads in a query block, weighting them based on the head's importance: $c(A) = \sum_{u \in U} w_u \phi_u(\sum_{v\in A} m_u(v))$ where $w_u > 0$.
>
>
> **RTR3**: Thank you for pointing this out. We will fix this in our updated manuscript.
>
>
> **RTR4**: We tailored our analysis on lm-eval-harness datasets to align with the published H2 results that used LLaMA-1 models. For additional datasets, we employed LLaMA-2 models. As models become more proficient in utilizing the input context, we expect the mixture functions $f$ and $c$ in Eq 4, which are based on key embeddings and accumulated attention scores, to more effectively capture key-space representativeness and context relevance.
>
>
> **RTR5**: We benchmark against baselines suitable for streaming. FastGen's need for targeted profiling across all attention heads over the entire sequence makes it unsuitable for our setting. Similarly, gisting, which trains an LLM to condense input prompts into gist tokens, is incompatible with our setting where no further finetuning is needed.
>
> **Q1**: $m_u(v)$ represents the total/accumulated attention the key $v$ receives from the observed input queries.
>
>
> **Q2**: Our current formulation (Eq 4) can be easily applied to GQA. See our response to RTR2.
>
>
> **Q3**: Fig 3, 6, and 7 use online BumbleBee algorithm, while Fig. 8, based on the LongBench dataset, uses the offline BumbleBee variant. We are working on LongChat results using the log-based concave function and will post them soon.

---

> > ### Comment · Reviewer_wh5U · 2024-06-06
> >
> > Thank you for the response. In conjunction with the additional results (Tables 1-4) provided in response to the other reviews, most of my concerns have been addressed. I will raise my score accordingly.
> >
> > It would be helpful to discuss asymptotic compute and memory costs in Section 5. In addition, I think a more intuitive justification for the two specific submodular functions proposed here—and why they complement each other—would be useful to include in the main paper. Readers may also have questions about generalization to different tasks and experimental settings, especially as Fig. 3 suggests that performance can be fairly sensitive to $\lambda$ and model size, so the new ablation experiments could be helpful in the appendix.

---

> > > ### Author Response · Authors · 2024-06-06
> > > **Thank you**
> > >
> > > We thank the reviewer for carefully reviewing our rebuttal and taking the time to revise their assessment. We are pleased that our response addressed the main concerns. Your detailed feedback and suggestions throughout the entire process have been invaluable, and we will incorporate them into our updated manuscript.
> > >
> > >
> > > Regarding Q3, we provide the LongChat results using the log-based concave function shown in the table at https://imgur.com/a/NTon3cO and will update the corresponding table in our revised manuscript.

---

### Official Review · Reviewer_v71g · 2024-05-12

**Rating:** 7
**Confidence:** 4
**Ethics Flag:** 1

**Summary:**

This paper introduces a novel approach to manage the memory overhead of Transformer-based LLMs by summarizing the key-value (KV) cache dynamically. The proposed BumbleBee framework uses a mixture of submodular functions to optimize the selection of key-value pairs, balancing diversity and importance based on accumulated attention scores. This method enables efficient memory utilization without sacrificing accuracy, even with long context sequences. BumbleBee operates both offline and online, accommodating various operational phases of LLMs. The effectiveness of this approach is validated through experiments on multiple datasets, showing improved performance over existing techniques with similar context reduction ratios.

**Questions To Authors:**

1. Could the authors elaborate on the theoretical underpinnings of the optimization problem utilized in the BumbleBee framework? Specifically, are there any provable guarantees regarding the convergence rate or the quality of the summarization achieved through your submodular optimization approach?

2. The experimental section lists two different concave functions—log-based and power-based—for the submodular formulation in Tables 1 and 2. Could the authors clarify the rationale behind using these two distinct function choices? Additionally, how do these choices influence the model's performance and why were they not discussed earlier in the methodology section?

**Reasons To Accept:**

1. **Novelty**: The approach of using a submodular optimization framework to balance the diversity and importance of key-value pairs in the memory cache of Transformers is novel and addresses a significant challenge in handling long contexts with LLMs effectively.

2. **Technical Soundness**: The paper is technically sound with a clear explanation of the submodular function utilized.

3. **Experimental Validation**: The paper presents comprehensive experiments across multiple datasets and compares the BumbleBee framework with several baselines, showing improvements in reducing memory overhead while retaining or even improving model accuracy.

**Reasons To Reject:**

1. **Lack of Theoretical Analysis**: While the empirical results are promising, the paper lacks a theoretical analysis of the convergence properties or bounds of the submodular optimization framework, which would strengthen the validation of the method.

2. **Impact of Hyperparameters**: The paper provides a brief discussion on the impact of the mixture weight hyperparameter $\lambda$, but it lacks a comprehensive analysis of other critical hyperparameters that may influence the framework's performance. Additionally, the study does not include ablation experiments, such as evaluations using only $f(A)$ or only $c(A)$, which could offer insights into the individual contributions of different components of the submodular function to the overall effectiveness of the BumbleBee framework.

---

> ### Author Rebuttal · Authors · 2024-05-31
>
> Thank you for your insightful feedback and for appreciating the importance of our work.
>
> **RTR1 & Q1**: Existing streaming submodular optimization algorithms like Sieve-Streaming (Badanidiyuru et al., 2014) and Sieve-Streaming++ (Kazemi et al., 2019) evaluate the final summary only after processing the entire data stream. However, real-time LLM decoding requires continuously maintaining optimal summaries, equivalent to generating the highest possible minimum-quality summary. Integrating these into transformers complicates theoretical proofs concerning performance and convergence due to the transformer's non-linear nature. Analyzing facility location function in a streaming setting is also challenging as the ground set changes with the stream.
>
> Line 11 in Alg. 2 is exact at each step - the reason, given the knowledge of the sequence observed so far at time t and running summary $S_{t-1}$ of fixed size $\tau_s$, it discards the item with least marginal gain, ensuring the resultant summary's function valuation is maximum among all $(\tau_s+1)$ candidate summaries.
>
>
> **RTR2**: Besides mixture weight, $\lambda$, our framework involves another hyperparameter: similarity measure in Eq 2. Since self-attention head similarities between query and key embeddings use dot products, we also use a dot-product-based similarity measure. Truncating negative entries to get non-negative similarity values (to preserve submodularity) worked best across three tasks (https://imgur.com/a/QR42zP0).
>
> Please find ablation results at https://imgur.com/a/vKUX9QT. Using mixture function with non-zero $\lambda$ to capture both representativeness and token importance performs the best.
>
>
> **Q2**: The choice of the concave function $\phi_u$ used to define the feature-based function is another hyperparameter. We experimented with two functions and found both perform equally well except on RTE dataset. These functions were selected for their similar diminishing returns behavior to the facility location function to ensure one did not dominate the other. The chosen $\phi_u$ determines how the marginal gain of adding a new key diminishes in the context of the current summary. For larger models with grouped-query attention, one can selectively choose the concave function $\phi_u$ in Eq. 3 to control the degree of diminishing returns for different query heads in a query block, weighting them based on importance such that $c(A) = \sum_{u \in U} w_u \phi_u(\sum_{v\in A} m_u(v))$ where $w_u > 0$.

---

> > ### Comment · Reviewer_v71g · 2024-06-06
> >
> > Thank you for the response. Your response has addressed most of my concerns. Good luck!

---

> > > ### Author Response · Authors · 2024-06-06
> > > **Thank you**
> > >
> > > We thank the reviewer for carefully reviewing our rebuttal. We are pleased that our response addressed the main concerns and sincerely appreciate your detailed feedback and suggestions throughout the review process.

---

### Decision · Program_Chairs · 2024-07-10

**Decision:**

Accept

**Comment:**

This paper presents a method to compress the KV cache in LLMs using a mixture of submodular functions to select a diverse and representative set of keys from the context. The authors have done a good job during rebuttal. After rebuttal, the paper received scores of 6777. All the reviewers are happy about the paper, commenting that (1) the proposed method is novel and interesting, (2) the paper is technically sound and well written, and (3) experimental results are comprehensive and sufficient. Therefore, the AC would like to recommend acceptance of the paper.